**Data Availability Statement:** The datasets are freely available for download at https://dhsprogram.com/data/available-datasets.cfm, as

# Female adolescents' reproductive health decision-making capacity and contraceptive use in sub-Saharan Africa: What does the future hold?

Bright Opoku Ahinkorah[1], John Elvis Hagan, Jr.[2,3], Abdul-Aziz Seidu[4,5]*, Francis Sambah[2], Faustina Adoboi[6], Thomas Schack[3], Eugene Budu[4]

1 The Australian Centre for Public and Population Health Research [ACPPHR], Faculty of Health, University of Technology Sydney, Sydney, Australia, 2 Department of Health, Physical Education, and Recreation, University of Cape Coast, Cape Coast, Ghana, 3 Neurocognition and Action-Biomechanics-Research Group, Faculty of Psychology and Sport Sciences, Bielefeld University, Bielefeld, Germany, 4 Department of Population and Health, University of Cape Coast, Cape Coast, Ghana, 5 College of Public Health, Medical and Veterinary Sciences, James Cook University, Townsville, Queensland, Australia, 6 Cape Coast Nursing and Midwifery Training College, Cape Coast, Ghana

* abdul-aziz.seidu@stu.ucc.edu.gh

## Abstract

### Introduction

Given the social, economic, and health consequences of early parenthood, unintended pregnancy, and the risks of HIV infection and subsequent transmission, there is an urgent need to understand how adolescents make sexual and reproductive decisions regarding contraceptive use. This study sought to assess the association between female adolescents' reproductive health decision-making capacity and their contraceptive usage.

### Materials and methods

Data was obtained from pooled current Demographic and Health Surveys (DHS) conducted in 32 countries in sub-Saharan Africa (SSA). The unit of analysis for this study was adolescents in sexual unions [n = 15,858]. Bivariate and multivariable analyses were conducted using Pearson chi-square tests and binary logistic regression respectively. All analyses were performed using STATA version 14.2. Results were presented using Odds Ratios [OR] and adjusted Odds Ratios [AOR]. Statistical significance was set at p<0.05.

### Results

The results showed that 68.66% of adolescents in SSA had the capacity to make reproductive health decisions. The overall prevalence of contraceptive use was 18.87%, ranging from 1.84% in Chad to 45.75% in Zimbabwe. Adolescents who had the capacity to take reproductive health decisions had higher odds of using contraceptives [AOR = 1.47; CI = 1.31–1.65, p < 0.001]. The odds of contraceptive use among female adolescents increased with age, with those aged 19 years having the highest likelihood of using contraceptives

described in the methods section of the manuscript.

**Funding:** The author(s) received no specific funding for this work.

**Competing interests:** The authors have declared that no competing interests exist.

[AOR = 3.12; CI = 2.27–34.29, p < 0.001]. Further, the higher the level of education, the more likely female adolescents will use contraceptives, and this was more predominant among those with secondary/higher education [AOR = 2.50; CI = 2.11–2.96, p < 0.001]. Female adolescents who were cohabiting had higher odds of using contraceptives, compared to those who were married [AOR = 1.69; CI = 1.47–1.95, p < 0.001]. The odds of contraceptive use was highest among female adolescents from the richest wealth quintile, compared to those from the poorest wealth quintile [AOR = 1.65; CI = 1.35–2.01, p<0.001]. Conversely, female adolescents in rural areas were less likely to use contraceptives, compared to those in urban areas [AOR = 0.78; CI = 0.69–0.89, p < 0.001].

## Conclusion

The use of general and modern contraceptives among adolescents in SSA remains low. Therefore, there is a need to strengthen existing efforts on contraceptives usage among adolescents in SSA. This goal can be achieved by empowering these young females, particularly those in the rural areas where the level of literacy is very low to take positive reproductive health decisions to prevent unintended teenage pregnancy, HIV/AIDs and other sexually transmitted infections. This approach would help reduce maternal mortality and early childbirth in studied SSA countries.

## Introduction

Empirical data from sub-Saharan Africa (SSA) suggest that a high proportion of 15- to 19-year-olds are sexually active and at risk of contracting HIV, other Sexually Transmitted Infections (STIs), or unplanned pregnancy because of lack of contraceptive use [1,2]. Although, recent statistics show that contraceptive use among young women in SSA has improved along with global trends [3,4], young women in these age brackets disproportionately use contraceptives, especially short-term methods (e.g., condoms, [4]). Given the heterogeneous nature of the SSA region, sub-regional variations exist in young women's contraceptive use. Generally, among women of reproductive age (i.e., 15–49), reported modern contraceptive use between 2005 and 2015 is highest (54.3%) in Southern Africa (i.e., Botswana, Lesotho, Namibia, South Africa, and Swaziland), followed by Northern Africa (i.e., Algeria, Egypt, Libya, Morocco, Sudan, and Tunisia). The moderate to lowest trends are recorded in Eastern (27.2%), Western (15%), and Central African (12%) countries [5]. According to Bundy et al. [6], it is possible that these developments could be similar for young populations across the region. This assertion explains why, on the average, 26–30% of presently married women aged 15–24 and 40–41% of recently unmarried women aged 15–24 report an unmet need for family planning in the West/Central and East/South regions of Africa [7].

Scholarly information reveals that empowering women is generally seen to be critical for effective implementation of family planning and reduction of HIV transmission [8–10]. Empowered women are often associated with the desire for fewer children, readily access more health services, avoid risky sexual behaviours, effectively strategize their births [10], and utilize condoms as HIV and STI prevention method [11,12]. Therefore, enhanced health and contraceptive accessibility and usage may empower women [13]. For instance, women who accessed and utilized diaphragms applauded their recent aptitude to demonstrate safer sexual behaviour in Kenya [14]. Similarly, Blanc [15] showed that the steadiness of power in sexual

relations has association with usage of health services and reproductive health outcomes. Despite a sizeable body of literature demonstrating the association between various dimensions of women empowerment on contraceptive usage, only few studies have explicitly measured the association between women reproductive health decision-making capacity and contraceptive use in SSA [16–19]. For example, Bankole and Singh [16] found that contraceptive usage was more pronounced when both couples decided to halt childbearing and reported lowest when both partners desired more children. Do and Kurimoto [17] later showed a positive association between women empowerment score and contraceptive use in selected countries (i.e., Ghana, Namibia, Uganda, Zambia, and Zimbabwe).

Similarly, findings reported by Darteh et al. [18] and Seidu et al. [19] on reproductive health decision making were very limited in scope (i.e., data from one country) and cannot be used for multi-country comparisons and generalizations across SSA. Even more compelling for additional research is that, in many SSA countries, women's right to health, including sexuality, has been infringed because of socio-cultural barriers (e.g., most decisions taken by men) [20–22].

Given the sparse literature on reproductive health decision-making among adolescents in SSA, the current study offers a baseline data for designing strategic programmes and policy that better promote female adolescents' health decision-making capacity and contraceptive use behaviour for their later healthy sexual and reproductive lives [23,24]. There is a general call for more research, policy, and programming attention that meet the reproductive health needs of young adult African women [4,25,26]. Therefore, the aim of this multi-country study was to assess the association between female adolescents' reproductive health decision-making capacity and their contraceptive usage. We anticipate that female adolescents' reproductive health decision-making capacity would shape their socio-cultural norms (e.g., subjective gender role orientations) and positively be related to contraceptive use.

## Materials and methods

### Data source

The study used pooled data from current DHS conducted from January 1, 2010 and December 31, 2018 in 32 countries in SSA. Table 1 provides a full list of the countries included in the survey and their survey years. The 32 countries were used in the study because they had complete data on the variables of interest for this study. The study included the countries under the DHS program in order to provide a holistic and in-depth evidence of the relationship between reproductive health decision-making capacity and contraceptive use in SSA. DHS is a nationwide survey collected every five-year period across low- and middle-income countries. The survey is representative of each of these countries. The survey targets core maternal and child health indicators such as unintended pregnancy, contraceptive use, skilled birth attendance, immunisation among under-fives, and intimate partner violence. Stratified dual-stage sampling approach was employed. The same questions were posed to all women, making it feasible for a multi-country comparisons. Selection of clusters (i.e., enumeration areas [EAs]) was the first step in the sampling process, followed by systematic household sampling within the selected EAs. The population for the study was obtained from adolescents aged 15–19 from whom data were collected during the current individual surveys for each country. For the purpose of this study, only adolescents (15–19 years) in sexual unions (marriage and cohabitation) who had complete cases on all the variables of interest were used (N = 15,858). Detailed description of the study sample is shown in Table 1. We relied on the Strengthening the Reporting of Observational Studies in Epidemiology' (STROBE) statement in conducting this study and writing the manuscript.

**Table 1. Description of the study sample (weighted).**

| Country | Survey Year | Adolescents (N) | Adolescents in sexual union (N) | Adolescents in sexual union (%) |
|---|---|---|---|---|
| Angola | 2015–16 | 3363 | 618 | 3.90 |
| Benin | 2017–18 | 3335 | 276 | 1.74 |
| Burkina Faso | 2010 | 3349 | 1,038 | 6.55 |
| Burundi | 2016–17 | 3968 | 229 | 1.44 |
| Cameroon | 2011 | 3579 | 370 | 2.33 |
| Chad | 2014–15 | 3874 | 482 | 3.04 |
| Comoros | 2012 | 1291 | 184 | 1.16 |
| Congo | 2011–12 | 2163 | 409 | 2.58 |
| Congo DR. | 2013–14 | 3980 | 851 | 5.36 |
| Côte d'Ivoire | 2011–12 | 1995 | 405 | 2.56 |
| Ethiopia | 2016 | 3498 | 594 | 3.75 |
| Gabon | 2012 | 1833 | 219 | 1.38 |
| Gambia | 2013 | 2461 | 508 | 3.20 |
| Ghana | 2014 | 1756 | 99 | 0.62 |
| Guinea | 2018 | 2561 | 704 | 4.44 |
| Kenya | 2014 | 2861 | 292 | 1.84 |
| Lesotho | 2014 | 1542 | 57 | 0.36 |
| Liberia | 2013 | 1914 | 289 | 1.83 |
| Malawi | 2015–16 | 5273 | 1,236 | 7.80 |
| Mali | 2018 | 2209 | 888 | 5.60 |
| Mozambique | 2015 | 1554 | 613 | 3.87 |
| Namibia | 2013 | 1857 | 96 | 0.60 |
| Niger | 2012 | 1901 | 1,100 | 6.94 |
| Nigeria | 2018 | 8423 | 810 | 5.11 |
| Rwanda | 2014–15 | 2779 | 84 | 0.53 |
| Senegal | 2010–11 | 3604 | 748 | 4.72 |
| Sierra Leone | 2013 | 4050 | 670 | 4.23 |
| South Africa | 2016 | 1505 | 42 | 0.27 |
| Togo | 2013–14 | 1732 | 210 | 1.32 |
| Uganda | 2016 | 4276 | 854 | 5.38 |
| Zambia | 2013–14 | 3686 | 441 | 2.78 |
| Zimbabwe | 2015 | 2156 | 438 | 2.76 |
| Total | | | 15,858 | 100.0 |

Source: Authors' computations

## Study variables

**Dependent variable.** The dependent variable in this study was "contraceptive use" which was derived from 'current contraceptive method'. The responses were coded 0 = "No method", 1 = "folkloric method", 2 = "traditional method," and 3 = "modern method". The existing DHS variable excluded women who were pregnant and those who had never had sex. The modern methods included female sterilization, intrauterine contraceptive device (IUD), contraceptive injection, contraceptive implants (Norplant), contraceptive pill, condoms, emergency contraception, standard day method (SDM), vaginal methods (foam, jelly, suppository), lactational amenorrhea method (LAM), country-specific modern methods, and respondent-mentioned other modern contraceptive methods (e.g., cervical cap, contraceptive sponge). Periodic abstinence (rhythm, calendar method), withdrawal (coitus interruptus), and country-

specific traditional methods of proven effectiveness were considered as traditional methods while locally described methods and spiritual methods (e.g., herbs, amulets, gris-gris) of unproven effectiveness were the folkloric methods. To obtain a binary outcome, all respondents who said they used 'no method' were put in one category and were given the code "0 = No" whereas those who were using either folkloric, traditional, or modern method were also put into one category and given the code "1 = Yes."

**Explanatory variables.**   The main explanatory variable was reproductive health decision-making capacity. Two derivative variables that focused on decision-making on sexual intercourse and condom use were used to generate this variable. For decision-making on sexual intercourse, female adolescents were asked whether they can refuse their partners sex while for condom use, female adolescents were asked whether they can ask their partners to use condoms during sexual intercourse. Reproductive health decision-making capacity was then generated from the combination of the decision-making on sexual intercourse and condom use variables. Following previous studies [18,19] on reproductive health decision-making capacity, the original response category of these variables (1 = Yes, 2 = No and 3 = Don't know/Not sure) were re-categorised, whereby "No and Don't know/Not sure" were recoded as "No" and recoded as 0, with "Yes" recoded as 1. Responses coded as "0" represented female adolescents' inability to make reproductive health decision whilst those coded as "1" were labelled as those capable of making reproductive health decisions.

Apart from the main explanatory variable, eight other variables were considered in the study as covariates. These variables were age, place of residence, wealth quintile, employment status, educational level, marital status, age at first sex, and survey country. Again, apart from survey country, these variables were not determined a priori; instead, they were selected based on their significant associations with the outcome variable—contraceptives use as reported in previous studies [27,28]. Four of the covariates were recoded to make them meaningful for analysis and interpretation. Marital status was recoded into "cohabiting (1)" and "married (2)." Employment status was captured as "not working (0)" and "working (1)." Educational level was recoded as "no education (0)," 'primary (1)," and "secondary/higher (2)." Age at first sex was coded as "less than 16 years (1)" and "16–19 years (2)."

## Statistical analyses

The analysis begun with a computation of contraceptive prevalence and proportion of female adolescents who could refuse a partner sex, ask a partner to use condom during sex, and those who had the capacity to make reproductive health decisions in the 32 SSA countries. The syntax "metaprop" in STATA version 14.0 (StataCorp, College Station, TX, USA) was used to generate forest plots for each of these indicators. Each forest plot showed the prevalence of an indicator in individual countries and its corresponding weight, as well as the pooled prevalence in all the countries and its associated 95% confidence intervals (CI). A test of heterogeneity of the DHS data obtained for the different countries showed a high level of inconsistency ($I^2 > 50\%$), thereby warranting the use of a random effect model in all the meta-analysis (see Figs 1, 2, 3 and 4). Secondly, the datasets were appended and a total sample of 15,858 was generated. After appending, contraceptive prevalence across the socio-demographic characteristics with their significance levels and chi square values [$\chi^2$] were calculated. Multicollinearity test was also performed and with a mean VIF of 1.21, there was no evidence of multicollinearity between the variables. Using the explanatory variables which were significantly associated with contraceptive use ($p < 0.05$) among female adolescents from the chi-square test, a binary logistic regression analysis in a hierarchical order was performed. Model I looked at a bivariate analysis of the main independent

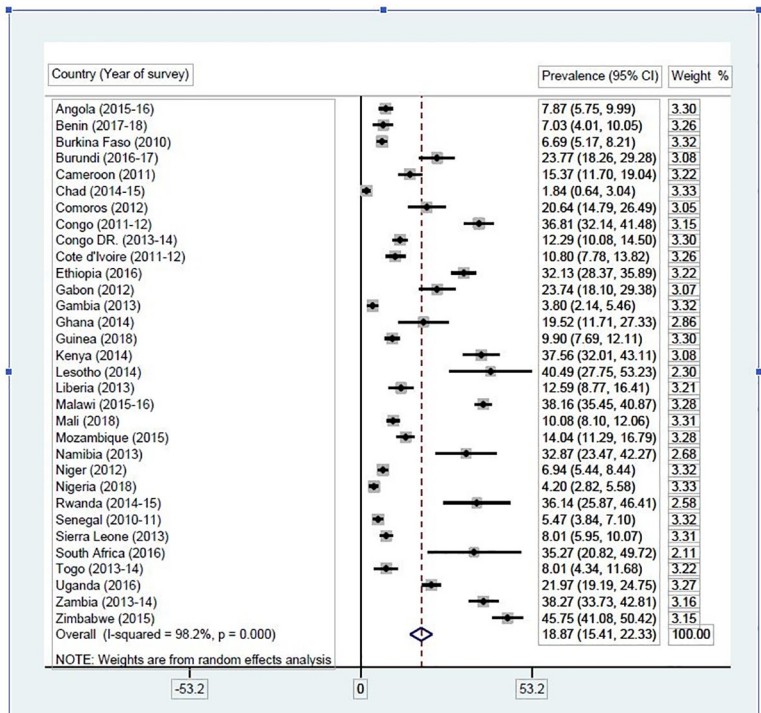

**Fig 1. Contraceptive prevalence among female adolescents in SSA.** Source: Authors' computations.

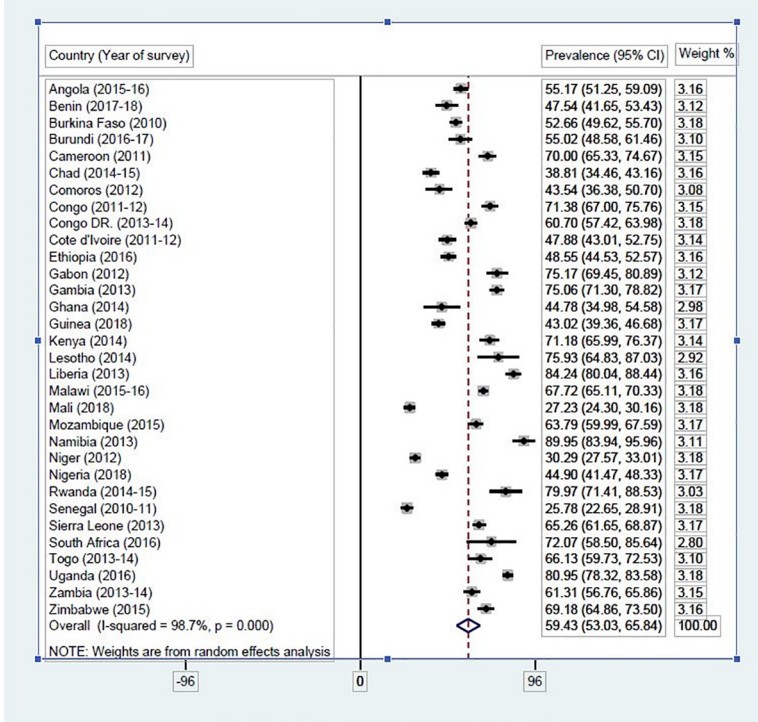

**Fig 2. Proportion of female adolescents in SSA who can refuse to have sex with a partner.** Source: Authors' computations.

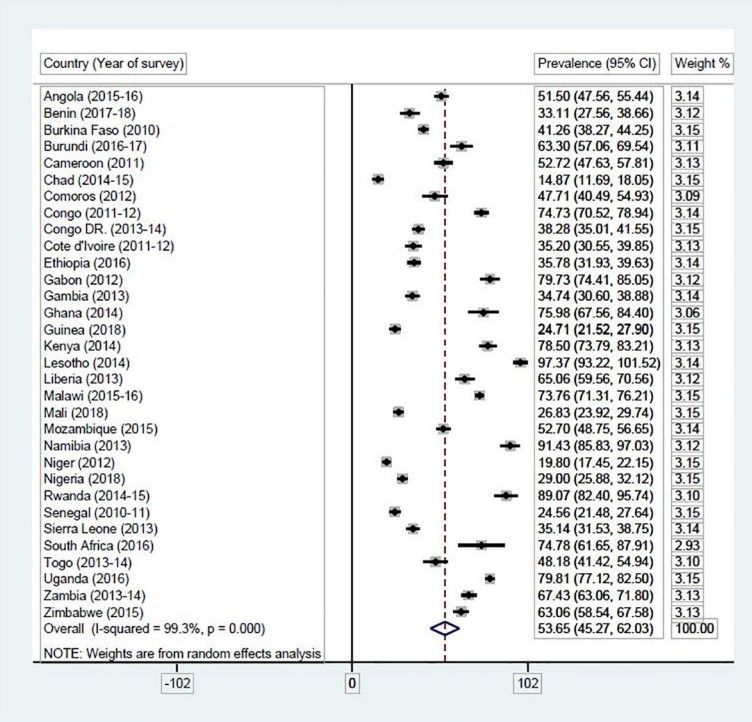

**Fig 3. Proportion of female adolescents in SSA who can ask their partners to use condom during sex.** Source: Authors' computations.

variable (reproductive health decision-making capacity) and the outcome variable (contraceptive use). Model II was a complete model comprising all the explanatory variables and the outcome variable (see Table 3). In line with research evidence that modern contraceptive methods are the most effective [29,30,31], a further analysis was done to examine the association between reproductive health decision-making capacity and modern contraceptive use (see Table 4). All frequency distributions were weighted using v005/1000000 while the survey [svy] command in STATA version 14.2 was used to adjust for the complex sampling structure of the data in the regression analyses. Missing values were treated by using complete cases for our analysis. Results for the regression analysis have been presented as Crude Odds Ratios (COR) and Adjusted Odds Ratios (AOR), with their corresponding 95% confidence intervals (CI) that signify precision and significance of the reported OR values. Statistical significance was set at p<0.05.

## Ethical approval

The DHS surveys obtain ethical clearance from the Ethics Committee of ORC Macro Inc. as well as Ethics Boards of partner organisations of the various countries such the Ministries of Health. During each of the surveys, either written or verbal consent was provided by the women. Since the data was not collected by the authors of this manuscript, official permission was sought from MEASURE DHS website and access to the data was provided upon the request that was assessed and approved on 3rd April, 2019. Data is available on https://dhsprogram.com/data/available-datasets.cfm.

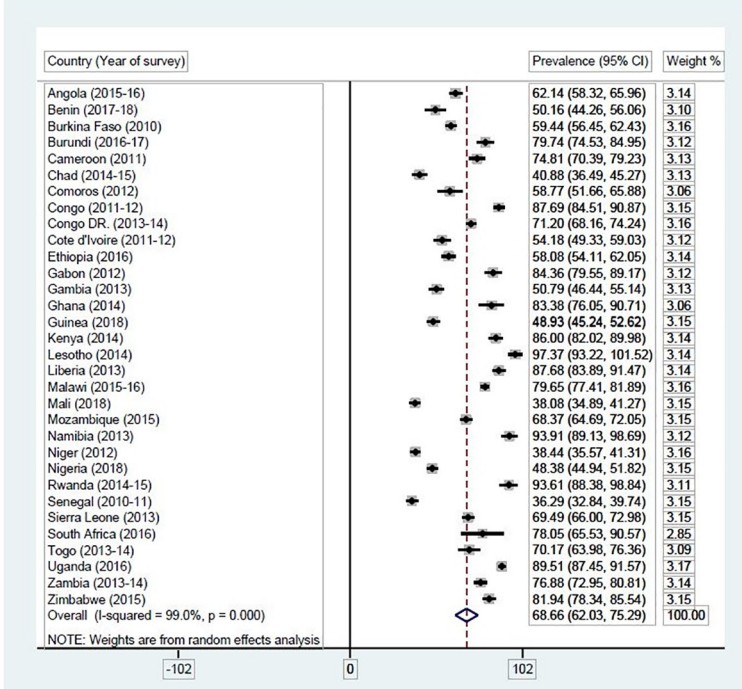

**Fig 4. Proportion of female adolescents in SSA who have the capacity to make reproductive health decisions.**
Source: Authors' computations.

## Results

### Percentage scores on the prevalence of contraceptive use within selected SSA countries

The prevalence of contraceptive use in each of the 32 SSA countries included in the study are presented in Fig 1. The overall prevalence of contraceptive use in SSA was 18.87% (95% CI: 15.41–22.33), ranging from 1.84% (95% CI: 0.64–3.04) in Chad to 45.75% (95% CI: 41.08–50.42) in Zimbabwe.

**Proportion of female adolescents who can refuse to have sex with a partner.** Fig 2 presents results on the proportion of female adolescents in SSA who could refuse have sex with their partners. The results indicate that 59.43% (95% CI = 53.03–65.84) of female adolescents could refuse sex to their partners in SSA. Majority of female adolescents who could refuse their partners sex were in Namibia [89.95% (95% CI = 83.94–95.96)] while the smallest proportion was in Senegal [25.78% (95% CI = 22.65–28.91)].

**Proportion of female adolescents who can ask their partners to use condom during sex.** Fig 3 presents results on the proportion of female adolescents in SSA who could ask their partners to use condom during sex. Overall, 53.65% (95% CI = 45.27–62.03) of female adolescents could ask their partners to use condom during sex in SSA. Out of this number, majority of them were in Namibia [91.43% (95% CI = 85.83–97.03)] while a few were in Niger [19.80% (95% CI = 17.45–22.15)].

**Proportion of female adolescents in SSA who have the capacity to make reproductive health decisions.** Fig 4 presents results on the proportion of female adolescents in SSA who have the capacity to make reproductive health decisions. Overall, 68.66% (95% CI: 62.03–75.29) of female adolescents in all the 32 SSA countries had the capacity to make reproductive

health decisions. Female adolescents in Senegal had the lowest reproductive health decision-making capacity [36.29% (95% CI: 32.84–39.74] while those in Lesotho had the highest proportion of reproductive health decision-making capacity [97.37% (95% CI: 93.22–101.52].

## Chi-square results on contraceptive use across socio-demographic characteristics

Table 2 provides a summary of the proportion of contraceptive use across reproductive health decision-making capacity and the selected socio-demographic characteristics. Nineteen-year-old female adolescents had the highest proportion of contraceptive use (21.8%) compared to those aged 15 (6.2%). Contraceptive use was high among female adolescents in urban areas (20.9%), compared to those in rural areas (14.4%). With wealth quintile, female adolescents from the richest wealth quintile had the highest proportion of contraceptive use (24.7%). There was no variation in contraceptive use in terms of employment status (i.e., not working-

**Table 2. Reproductive health decision-making capacity, socio-demographic characteristics, and contraceptive use among female adolescents in SSA.**

| Variables | Weighted N | Weighted % | Contraceptive use | |
|---|---|---|---|---|
| Reproductive health decision-making capacity [$\chi^2$ = 379.6, p<0.001] | | | N | % |
| Incapable | 5,887 | 37.1 | 512 | 8.6 |
| Capable | 9,970 | 62.9 | 2,021 | 20.4 |
| Age [$\chi^2$ = 257, p<0.001] | | | | |
| 15 | 771 | 4.9 | 49 | 6.2 |
| 16 | 1,551 | 9.8 | 153 | 9.7 |
| 17 | 2,798 | 17.6 | 385 | 13.9 |
| 18 | 5,337 | 33.7 | 775 | 14.5 |
| 19 | 5,401 | 34.1 | 1,171 | 21.8 |
| Place of residence [$\chi^2$ = 94.4, p<0.001] | | | | |
| Urban | 3,891 | 24.5 | 813 | 20.9 |
| Rural | 11,967 | 75.5 | 1,720 | 14.4 |
| Wealth quintile [$\chi^2$ = 141.9, p<0.001] | | | | |
| Poorest | 3,947 | 24.9 | 564 | 12.8 |
| Poorer | 3,927 | 24.8 | 584 | 15.0 |
| Middle | 3,472 | 21.9 | 503 | 15.3 |
| Richer | 2,778 | 17.5 | 470 | 18.2 |
| Richest | 1,734 | 10.9 | 412 | 24.7 |
| Employment status [$\chi^2$ = 0.003, p = 0.960] | | | | |
| Not working | 7,420 | 46.8 | 1,233 | 16.0 |
| Working | 8,438 | 53.2 | 1,300 | 16.0 |
| Marital status [$\chi^2$ = 156.1, p<0.001] | | | | |
| Married | 11,810 | 74.5 | 1,616 | 13.8 |
| Cohabiting | 4,048 | 25..5 | 917 | 22.1 |
| Educational level [$\chi^2$ = 877.6, p<0.001] | | | | |
| No Education | 6,021 | 38.0 | 319 | 5.3 |
| Primary | 6,100 | 38.5 | 1,277 | 20.6 |
| Secondary/higher | 3,737 | 23.6 | 937 | 25.9 |
| Age at first sex [$\chi^2$ = 0.3, p = 0.601] | | | | |
| Less than 16 years | 8,790 | 55.4 | 1,409 | 15.8 |
| 16–19 years | 7,068 | 44.6 | 1,124 | 16.1 |

Source: Authors' computations

16%, working-16%). Female adolescents who were cohabiting recorded the highest proportion of contraceptive use (22.1%), compared to those who were married (13.8%). Female adolescents with secondary/higher education (25.9%) and those whose age at first sex ranged between 16 and 19 years (16.1%) had the highest proportion of contraceptive use respectively. Apart from occupation and age at first sex, all the explanatory variables showed statistically significant associations with contraceptive use among female adolescents (p<0.05).

## Binary logistic regression analysis on female adolescents' reproductive health decision-making capacity and contraceptive use

Results in Table 3 on reproductive health decision-making capacity and contraceptive use showed that female adolescents who had the capacity to make reproductive health decisions were more likely to use contraceptives [OR = 2.70; CI = 2.44–3.00, $p < 0.001$], compared to those who did not have the capacity to make reproductive health decision, even after controlling for the effect of the covariates [AOR = 1.47; CI = 1.31–1.65, $p < 0.001$]. Compared to Chad, female adolescents in Zimbabwe had the highest odds of using contraceptives [AOR = 30.44; CI = 14.49–63.93, $p < 0.001$]. The odds of contraceptive use among female adolescents increased with age, with those aged 19 years having the highest likelihood of using contraceptives [AOR = 3.12; CI = 2.27–4.29, $p < 0.001$]. The results further showed that the higher the level of education, the more likely female adolescents will use contraceptives, and this was more predominant among those with secondary/higher education level of education [AOR = 2.50; CI = 2.11–2.96, $p < 0.001$].

Female adolescents who were cohabiting had higher odds of using contraceptives, compared to those who were married [AOR = 1.69; CI = 1.47–1.95, $p < 0.001$]. The odds of contraceptive use was highest among female adolescents from the richest wealth quintile, compared to those from the poorest wealth quintile [AOR = 1.65; CI = 1.35–2.01, $p < 0.001$]. Conversely, female adolescents in rural areas were less likely to use contraceptives, compared to those in urban areas [AOR = 0.78; CI = 0.69–0.89, $p < 0.001$] (see Table 3).

## Binary logistic regression analysis on female adolescents' reproductive health decision-making capacity and modern contraceptive use

Results in Table 4 on reproductive health decision-making capacity and modern contraceptive use showed that female adolescents who had the capacity to make reproductive health decisions were more likely to use modern contraceptives [OR = 2.59; CI = 2.33–2.89], $p < 0.001$], compared to those who did not have the capacity to make reproductive health decision, even after controlling for the effect of the covariates [AOR = 1.43; CI = 1.26–1.61, $p < 0.001$]. Compared to Chad, the highest odds of modern contraceptives use was among female adolescents in Zimbabwe [AOR = 39.58; CI = 17.06–91.84, $p < 0.001$]. The likelihood of modern contraceptive use among female adolescents increased with age, with those aged 19 years having the highest likelihood of using contraceptives [AOR = 2.77; CI = 1.98–3.87, $p < 0.001$]. The odds of modern contraceptives usage was low among female adolescents in rural areas, compared to those in urban areas [AOR = 0.73; CI = 0.63–0.84, $p < 0.001$].

The results further showed that the higher the level of education, the more likely female adolescents will use modern contraceptives, and this was more predominant among individuals with secondary/higher level of education [AOR = 2.51; CI = 2.09–3.00, p<0.001]. Female adolescents who were cohabiting were more likely to use modern contraceptives, compared to those who were married [AOR = 1.60; CI = 1.37–1.86, $p < 0.001$]. The odds of modern contraceptive use was highest among female adolescents from the richest wealth quintile, compared to those from the poorest wealth quintile [AOR = 1.55; CI = 1.26–1.91], $p < 0.001$] (see Table 4).

**Table 3. Binary logistic regression analysis on female adolescents' reproductive health decision-making capacity and contraceptive use.**

| Variables | Model I COR [95%CI] | Model II AOR [95%CI] |
|---|---|---|
| **Reproductive health decision-making capacity** | | |
| Incapable | Ref | Ref |
| Capable | 2.70***[2.44–3.00] | 1.47***[1.31–1.65] |
| **Survey country** | | |
| Angola | | 1.82 [0.83–3.99] |
| Burkina Faso | | 4.00*** [1.90–8.45] |
| Benin | | 3.28*** [1.39–7.78] |
| Burundi | | 8.34*** [3.89–18.28] |
| Congo DR. | | 3.89*** [1.86–8.15] |
| Congo | | 14.28*** [6.75–30.21] |
| Côte d'Ivoire | | 5.02** [1.65–29.80] |
| Cameroon | | 6.25*** [2.93–13.35] |
| Ethiopia | | 16.78*** [8.10–34.77] |
| Gabon | | 8.38*** [3.86–18.18] |
| Ghana | | 6.72*** [2.83–15.98] |
| Gambia | | 1.37 [0.56–3.35] |
| Guinea | | 5.69*** [2.67–12.09] |
| Kenya | | 15.38***[7.28–32.47] |
| Comoros | | 8.10***[3.66–17.92] |
| Liberia | | 3.14**[1.42–6.97] |
| Lesotho | | 15.20*** [6.06–38.05] |
| Mali | | 6.40***[3.06–13.41] |
| Malawi | | 27.09*** [13.22–55.53] |
| Mozambique | | 5.84***[2.76–12.37] |
| Nigeria | | 2.83**[1.29–6.18] |
| Niger | | 6.76***[3.23–14.15] |
| Namibia | | 13.92***[5.99–32.34] |
| Rwanda | | 12.77***[5.43–40.05] |
| Sierra Leone | | 3.76*** [1.76–8.02] |
| Senegal | | 5.53**[1.3–7.66] |
| Chad | | Ref |
| Togo | | 3.40** [1.45–7.97] |
| Uganda | | 6.88*** [3.30–14.34] |
| South Africa | | 23.59*** [8.70–63.98] |
| Zambia | | 27.73*** [13.29–57.85] |
| Zimbabwe | | 30.44***[14.49–63.93] |
| **Age** | | |
| 15 | | Ref |
| 16 | | 1.46*[1.02–2.07] |
| 17 | | 2.19***[1.57–3.04] |
| 18 | | 2.25***[1.64–3.10] |
| 19 | | 3.12***[2.27–4.29] |
| **Place of residence** | | |
| Urban | | Ref |
| Rural | | 0.78***[0.69–0.89] |
| **Educational level** | | |

*(Continued)*

**Table 3.** (Continued)

| Variables | Model I COR [95%CI] | Model II AOR [95%CI] |
|---|---|---|
| No Education | | Ref |
| Primary | | 1.98***[1.70–2.29] |
| Secondary/Higher | | 2.50***[2.11–2.96] |
| **Marital status** | | |
| Married | | Ref |
| Cohabiting | | 1.69***[1.47–1.95] |
| **Wealth quintile** | | |
| Poorest | | Ref |
| Poorer | | 1.20**[1.05–1.37] |
| Middle | | 1.21*[1.05–1.37] |
| Richer | | 1.35***[1.14–1.59] |
| Richest | | 1.65***[1.35–2.01] |
| N | 15,858 | 15,858 |
| pseudo $R^2$ | 0.029 | 0.178 |

* p<0.05

** p<0.01

*** p<0.001; Ref = Reference, CI = Confidence Intervals, COR = Crude Odds Ratio, AOR = Adjusted Odds Ratio

Source: Authors' computations

## Discussion

This multi-country study assessed the association between female adolescents' reproductive health decision-making capacity and their contraceptive usage in 32 SSA countries. The study was critical since females' ability to take decisive reproductive health decisions including choices in contraceptive use (i.e., condom use) can lead to good reproductive health [19] through the prevention of certain STIs (e.g., HIV and Hepatitis B) [26]. Generally, 68.66% of studied females across the 32 countries had the capacity (i.e., empowerment) to make reproductive health decisions, a finding that is similar but smaller than the 69.3% capacity found by Darteh, Dickson and Doku [32]. This variation might be due to differences in the study populations and the number of countries these studies were based on. We found that female adolescents who had the capacity to make reproductive health decisions were more likely to use contraceptives. This pattern does not necessarily imply that females who had no capacity (i.e., empowerment) had no priority of contraceptive use. Some scholars have found that females' restrictive roles in domestic matters including reproductive health issues in many SSA countries (e.g., Ghana, Kenya, Tanzania, Nigeria, Uganda, and South Africa) rest with men [33], a reason that could derail them from using contraceptive.

Female adolescents from Zimbabwe had the highest odds of contraceptive usage, compared to Angola. This result is not surprising because Zimbabwe, together with other southern African states (e.g., Namibia) have one of the most effective family planning programs in SSA, with lowest unmet need for contraception among adolescent women in Africa [34]. Other plausible reasons might be that Zimbabwe's positive success in female adolescents' contraceptive usage rest with their high literacy rate, suggesting that their adolescents might have capacity in many other areas of life such as socio-cultural and familial empowerment [35], compared to young females in the other 31 SSA countries. What may also be accounting for the lower odds of using contraceptives in other SSA countries could be ascribed to the low

**Table 4. Binary logistic regression analysis on female adolescents' reproductive health decision-making capacity and modern contraceptive use.**

| Variables | Model I COR [95%CI] | Model II AOR [95%CI] |
|---|---|---|
| **Reproductive health decision-making capacity** | | |
| Incapable | Ref | Ref |
| Capable | 2.59*** [2.33–2.89] | 1.43*** [1.26–1.61] |
| **Survey country** | | |
| Angola | | 2.27 [0.93–5.50] |
| Burkina Faso | | 5.03* [2.16–11.73] |
| Benin | | 3.63** [1.37–9.60] |
| Burundi | | 9.52** [3.92–23.17] |
| Congo DR. | | 1.78 [0.74–4.29] |
| Congo | | 9.08*** [3.83–21.48] |
| Côte d'Ivoire | | 3.89** [1.55–9.76] |
| Cameroon | | 6.06*** [2.54–14.44] |
| Ethiopia | | 21.96*** [9.60–50.27] |
| Gabon | | 7.20*** [2.97–17.46] |
| Ghana | | 8.16*** [3.11–21.42] |
| Gambia | | 1.38 [0.50–3.83] |
| Guinea | | 7.25*** [3.09–16.98] |
| Kenya | | 18.47*** [7.91–43.13] |
| Comoros | | 7.16*** [2.88–17.80] |
| Liberia | | 4.36*** [1.78–10.64] |
| Lesotho | | 20.58*** [7.57–55.99] |
| Mali | | 7.68*** [3.32–17.76] |
| Malawi | | 35.71*** [15.72–81.09] |
| Mozambique | | 7.57*** [3.24–17.73] |
| Nigeria | | 2.57* [1.04–6.33] |
| Niger | | 7.64*** [3.30–17.69] |
| Namibia | | 17.49*** [6.85–44.68] |
| Rwanda | | 16.18*** [6.25–41.86] |
| Sierra Leone | | 5.09*** [2.17–11.96] |
| Senegal | | 4.13** [1.72–9.88] |
| Chad | | Ref |
| Togo | | 4.10** [1.58–10.66] |
| Uganda | | 8.80*** [3.81–20.36] |
| South Africa | | 31.80*** [10.85–93.25] |
| Zambia | | 36.17*** [15.68–83.40] |
| Zimbabwe | | 39.58*** [17.06–91.84] |
| **Age** | | |
| 15 | | Ref |
| 16 | | 1.37 [0.94–1.99] |
| 17 | | 1.98*** [1.40–2.80] |
| 18 | | 1.98*** [1.42–2.77] |
| 19 | | 2.77*** [1.98–3.87] |
| **Place of residence** | | |
| Urban | | Ref |
| Rural | | 0.73*** [0.63–0.83] |
| **Educational level** | | |

*(Continued)*

**Table 4.** (Continued)

| Variables | Model I COR [95%CI] | Model II AOR [95%CI] |
|---|---|---|
| No Education | | Ref |
| Primary | | 2.01***[1.71–2.36] |
| Secondary/Higher | | 2.51***[2.09–3.00] |
| **Marital status** | | |
| Married | | Ref |
| Cohabiting | | 1.60***[1.37–1.86] |
| **Wealth quintile** | | |
| Poorest | | Ref |
| Poorer | | 1.17* [1.02–1.36] |
| Middle | | 1.20*[1.03–1.40] |
| Richer | | 1.32**[1.11–1.56] |
| Richest | | 1.55***[1.26–1.91] |
| *N* | 15,858 | 15,858 |
| pseudo $R^2$ | 0.026 | 0.188 |

* p<0.05

** p<0.01

*** p<0.001; Ref = Reference, CI = Confidence Intervals, COR = Crude Odds Ratio, AOR = Adjusted Odds Ratio

Source: Authors' computations

acceptance and high cultural resistance to family planning. According to Caldwell and Caldwell [36], the social, financial, and strict kingship norms and values devoted to children and family in the region are also contributory toward the application of contraceptives.

The present study recorded an age progression in contraceptive usage, with those aged 19 years having the highest likelihood of using contraceptives. This finding corroborates previous studies in low-and middle-income countries like Pakistan [37], Nepal [38], and Ghana [19], where females' reproductive health decision-making capacity increases with age as their other dimensions of empowerment increases. Female adolescents in rural dwellings were less likely to use contraceptives, compared to their urban counterparts. This disparity in rural-urban contraceptive use could be due to limited and lack of access to healthcare services in rural communities, and even where they exist, socio-cultural limitations may hinder usage [39]. This assertion reinforces the rural-urban inequities in terms of essential services like reproductive health (i.e., family planning services) for female adolescents. This finding can best be appreciated when a thoughtful reflection is done on how health facilities, and key human and other resources are skewed in favour of urban settings in most countries across SSA [40]. African governments and non-governmental organisations could roll out reproductive health services, such as the adoption and replication of Community Health Programme Services (CHPS) concept across the sub-region to grant quality healthcare access to rural dwellers.

The higher the level of education, the more likely some female adolescents will use contraceptives, predominantly among those with secondary/higher education. Specifically, the odds of contraceptive use was higher for those with secondary and/or higher education, compared with those with none, a finding similar to previous studies in Ghana and other low and middle-income countries [18,41]. This finding strongly shows that educating girls may be an effective tool for promoting reproductive health decision-making capacity and contraceptive use. Educated female adolescents may have better access to healthcare information and reproductive health literature, better ability to use quality healthcare services, and greater autonomy to

make decisions [42]. Other studies have also proven that education may impart feelings of self-worth and self-confidence, which are necessary for changing health behaviours and seeking health services for the first time [43].

The odds of contraceptive use was highest among female adolescents from the richest wealth quintile, compared to those from the poorest wealth quintile. This finding is consistent with previous studies in Africa [27] and Pakistan [44] that found women with high wealth index to have high affinity for contraceptive use. For example, Nyarko [45] explained that Ghanaian women who have high wealth status are more capable to take charge of their sexual and reproductive health matters (i.e., contraceptive use) than women of poorer wealth status. Drawing from the finding, rich women who are but few in Africa gain more repute, boost their confidence, and are self-reliant to independently decide and afford any contraceptive of their choice. Likewise, female adolescents from the poorest wealth quintile may need to be more sexually empowered than their richer counterparts to overcome the greatest barriers accounting for the utilization of health services, including contraceptives. African governments and gender empowerment NGOs within the sub-region should target much of their social interventions (i.e., pro-poor) programs toward adolescent females, as most women in Africa are below the poverty line [46]. Specific competency-based and life skills-based education training interventions should be implemented across the sub-region by governments to broaden the scope of female adolescents' reproductive health issues [47].

A statistically significant association between marital status and contraceptive use was found in the current study. Specifically, adolescents who were cohabiting had higher odds of using modern contraception, compared with those who were married. Similar to previous evidence [48–50], marital status is imperative and facilitates women's acceptability of sexual intercourse minus contraception in most traditional settings in SSA,. For instance, Gyan [51] explained that premarital sex and premarital childbearing is a source of stigma in many African communities and can negatively affect adolescent girls' sexual and reproductive health experiences. Due to this, some adolescent girls resort to the use of contraception to avoid this stigma.

## Practical implications

Female adolescents' ability to make reproductive health decisions was associated with contraceptive use. Nonetheless, it is imperative to recognize that female adolescents' involvement in positive reproductive decisions is strongly dependent on the shared or interactive influence of their age, rural-urban geographical location, educational status, and socio-economic status. For most parts of SSA, many decisions on managing sexual and reproductive health have come under serious scrutiny from a socio-cultural lens as female adolescents' vulnerability to unintended pregnancy, unsafe abortions, HIV and other STI's have increased. For example, a deep-rooted cultural barrier to contraceptive use in many parts of SSA could be the unfriendliness that many female adolescents experience when they visit maternal and child health clinics and other primary sources of contraceptives. Due to overt social dissatisfaction of premarital sexual activity, and the general lack of confidentiality and anonymity at these health centres, many young girls who attempt to procure contraceptives, are generally exposed to public ridicule, gossip, and negative attitudes from health personnel [52]. Therefore, female adolescents with lower reproductive health decision-making capacity could be at high risk of mistreatment from healthcare providers, either by getting poor-quality services or suspending health services all together [43].

Education, particularly secondary and/ or higher education, has repetitively been shown to be connected with a wide range of positive sexual and reproductive health outcomes such as

contraceptive use, age of marriage, number of births, and use of health services, albeit other proxies (e.g., socio-economic status). One systematic review on risk and protective factors for Adolescent Sexual and Reproductive Health in low- and middle-income countries found that the more years adolescents spend in school, the less likely they would ever engage in sex. These adolescents have greater chances of using modern contraceptives, compared to those who leave school early [53,54]. Other studies have proven that education may impart feelings of self-worth and self-confidence, which are necessary for changing health behaviour and seeking out health services for the first time [52]. Therefore, educational intervention programs that would promote the enrolment and retention of female adolescents in secondary schools and perhaps beyond are critical toward improving female adolescents' reproductive health decision-making capacity and their overall health.

Undoubtedly, though the DHS includes standard questions that allow multi-country comparisons, further in-depth qualitative enquiry and statistical linkages between different components of reproductive health decision-making and adolescents' likelihood to engage in sexual and reproductive health behaviours (e.g., onset of sexual debut, multiple sexual partners, unsafe abortions, unintended pregnancies) would be helpful to better understand the context-specific issues across SSA. Critical consideration must be given to context-specific institutional barriers (e.g., cultural norms and expectations for sex roles) to adolescents' sexuality such as pregnancy prevention, healthcare providers' attitudes, and lack of confidentiality, and their impact on adolescents' capacity to make decisions about sexual and reproductive health issues [17].

## Strengths and limitations

The main strength of the study is the use of nationally representative surveys that have been validated across the countries and as a result, make the findings of the current study valid and generalisable to other adolescents in SSA. Despite this strength, the current study is not without limitations. First, the DHS data presented in this study is limited to 32 SSA countries and, therefore, may not be applicable to other countries outside these geographical settings. Second, this study is linked to the cross-sectional DHS data. Given that the data were collected within a specific time frame, causal inferences or temporal relationship between studied variables cannot be ascertained. There is a probability that the similar observed or unobserved factors may impact both female adolescents' reproductive health decision-making capacity and contraceptive method used. For the purpose of robustness, some factors that are theoretically associated to contraceptive use but not to empowerment were statistically controlled; hence, full testing and adjusting for endogeneity is necessary but was beyond the scope of this study. Respondents needed to have answered "yes" to both "ability to refuse partners sex and ability to ask partner to use condom" in order to be considered to have 'reproductive health decision-making capacity. With this, women using another contraceptive method might find it strange or difficult to request that their partner, particularly in longer-term or 'more serious relationships', use a condom (i.e., double protection). Second, the outcome variable of contraceptive use is insufficiently nuanced to consider whether the adolescent has a need to avoid pregnancy. Also, the current study used the most recent DHS data, hence, variations in contraceptive accessibility and usage may have happened in the last few years. For example, countries like Zimbabwe and Namibia have been promoting long-standing family planning services in some of their districts, whereas Zambia has seen a drastic shift in funding from family planning to HIV prevention and management. While these alterations are not likely to have caused changes in contraceptive use within a specific time frame, possible long-term changes in these policy shifts may lead to increased usage of specific contraceptive methods (e.g., condom use) at the expense of others (i.e., IUD). Therefore, there may also be variations with the linkages

between reproductive health decision-making capacity and contraceptive use [42]. Due to the very sensitive nature of the sexuality themes under consideration, there is the possibility of social desirability concerns or recall bias among the study participants.

## Conclusions

Utilisation of contraceptives among adolescents remains low in SSA. This study revealed that reproductive health decision-making capacity, age, education, wealth status, marital status, and place of residence are associated with contraceptives use. Therefore, strengthening existing efforts in SSA on contraceptives use among adolescents by empowering adolescents to take reproductive health decisions, targeting younger adolescents, those in rural areas, those without formal education, those in the poor wealth status and those who are married and may want to delay pregnancy is critical. These interventions can help reduce teenage pregnancy, early childbirth, and maternal mortality and help in the achievement of SDG three. The current findings suggest that specific programs that aim to promote female adolescents' capacity to discuss sexual activity would be particularly helpful in SSA countries, especially Chad and Gambia that were identified with the lower odds of using contraceptives and where family planning services are limited. These programs should complement efforts to increase availability of contraceptives to meet the rising demand for modern methods due to serious child and maternal health issues in these countries.

## Acknowledgments

We are grateful to MEASURE DHS for granting us access to the data. We also acknowledge Mr. Ebenezer Agbaglo of the Department of English, University of Cape Coast, who thoroughly copy-edited this manuscript for language usage, spelling, and grammar.

## Author Contributions

**Conceptualization:** Bright Opoku Ahinkorah.

**Data curation:** Bright Opoku Ahinkorah, Abdul-Aziz Seidu.

**Formal analysis:** Bright Opoku Ahinkorah, Abdul-Aziz Seidu.

**Funding acquisition:** Bright Opoku Ahinkorah.

**Investigation:** Bright Opoku Ahinkorah, Abdul-Aziz Seidu.

**Methodology:** Bright Opoku Ahinkorah, Abdul-Aziz Seidu.

**Project administration:** Bright Opoku Ahinkorah, Abdul-Aziz Seidu.

**Resources:** Bright Opoku Ahinkorah, Abdul-Aziz Seidu.

**Software:** Bright Opoku Ahinkorah, Abdul-Aziz Seidu.

**Supervision:** Bright Opoku Ahinkorah, John Elvis Hagan, Jr., Abdul-Aziz Seidu.

**Validation:** Bright Opoku Ahinkorah, John Elvis Hagan, Jr., Abdul-Aziz Seidu.

**Visualization:** Bright Opoku Ahinkorah, Abdul-Aziz Seidu.

**Writing – original draft:** Bright Opoku Ahinkorah, John Elvis Hagan, Jr., Abdul-Aziz Seidu, Francis Sambah, Faustina Adoboi, Thomas Schack, Eugene Budu.

**Writing – review & editing:** Bright Opoku Ahinkorah, John Elvis Hagan, Jr., Abdul-Aziz Seidu, Francis Sambah, Faustina Adoboi, Thomas Schack, Eugene Budu.

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
