## [Decision Letter · Decision Letter 0]

9 Mar 2020

PONE-D-19-29601

Female adolescents’ reproductive health decision-making capacity and contraceptive use in sub-Saharan Africa: What does the future hold?

PLOS ONE

Dear Mr Seidu,

Thank you for submitting your manuscript to PLOS ONE. After careful consideration, we feel that it has merit but does not fully meet PLOS ONE’s publication criteria as it currently stands. Therefore, we invite you to submit a revised version of the manuscript that addresses the points raised during the review process.

While most reviewer comments are minor, there are some major issues with the methodology of the paper. Please pay careful attention to that section.

We would appreciate receiving your revised manuscript by Apr 23 2020 11:59PM. To enhance the reproducibility of your results, we recommend that if applicable you deposit your laboratory protocols in protocols.io, where a protocol can be assigned its own identifier (DOI) such that it can be cited independently in the future. For instructions see: http://journals.plos.org/plosone/s/submission-guidelines#loc-laboratory-protocols

We look forward to receiving your revised manuscript.

Kind regards,

Emily Vala-Haynes

Academic Editor

PLOS ONE

Journal Requirements:

2. Your ethics statement must appear in the Methods section of your manuscript. If your ethics statement is written in any section besides the Methods, please move it to the Methods section and delete it from any other section. Please also ensure that your ethics statement is included in your manuscript, as the ethics section of your online submission will not be published alongside your manuscript.

Reviewers' comments:

Reviewer's Responses to Questions

**Comments to the Author**

1. Is the manuscript technically sound, and do the data support the conclusions?

Reviewer #1: Yes

Reviewer #2: Yes

Reviewer #3: Partly

2. Has the statistical analysis been performed appropriately and rigorously? 

Reviewer #1: No

Reviewer #2: Yes

Reviewer #3: I Don't Know

3. Have the authors made all data underlying the findings in their manuscript fully available?

Reviewer #1: Yes

Reviewer #2: Yes

Reviewer #3: Yes

4. Is the manuscript presented in an intelligible fashion and written in standard English?

Reviewer #1: Yes

Reviewer #2: Yes

Reviewer #3: Yes

5. Review Comments to the Author

Reviewer #1: In the methodology, the author/authors suggested the use of Bivariate and multivariable analysis which include Pearson chi-square tests and binary logistic regression respectively, however, binary logistic regression model in most cases is misleading in interpreting the complex relationships of this nature. The authors need to demonstrate how they deal with existence of multicollinearity problems which obviously made the model deficiency.

I feel that the problem of multicollinearity cannot be undermined in this study because it can lead the authors into making incorrect conclusion about relationships between responsive and explanatory variables. Therefore, I would suggest that the authors should consider adopting Principle component analysis (PCA) in addition to regression approaches to avoid this problem. The PCA is a multivariate technique which help to understand the underlying data structure and to form a smaller number of uncorrelated new variables. In other words, PCA reduces number of predictors to avoid multicollinearity problem and it is highly recommended due to its ability to identify a small number of derived variables from a larger number of original variables in order to simplify the subsequent analysis of the data

Reviewer #2: I acknowledge the authors for a comprehensive and in terms of language, a well-written paper. The paper was easy to follow. Common, yet important limitations of studies of such types involving sexual and reproductive health (SRH) were detailed in the “Limitation” section. Few comments I have are as follows:

1. Important things to mention in the last part of the “Introduction” section always are to briefly mention similar studies, what they lacked (the research gaps) and which gaps the current study aims to fulfill. This would make clear for the readers how the study is different from other studies with similar research objective. This important aspect is seen to be missing in the “Introduction” part.

An example for it would be: In the material and method section, the authors mention to have followed previous studies (referenced 33 and 34) (Line no. 190). We see the topic of these two studies are similar to the current study, and study of such types are existent, however, the authors neither mention this study in the “Introduction” nor the gaps these studies had and the gaps that the current study attempts to cover. Another similar comment when the authors directly mention other two studies (reference 42, 43) (L.352) in the discussion and compare their results with them.

2. Although the study covered many aspects determining the reproductive decision making and contraceptive use, some interesting analysis could have been carried out additionally. I suggest a subgroup analysis grouping the countries with SRH or contraceptive intervention and the countries without. I believe that reproductive decision alone cannot facilitate contraceptive use (To explain, even if one has the reproductive decision making capacity but no availability of contraceptive, he/she would not be able to use contraceptives despite the will. On the other hand, if a person has very easy access to contraceptives, a slight decision making capacity would also be enough for contraceptive use).

3. L: 258-270: I suggest mentioning the CI of the given results. (CI across different countries).

Minor comments:

1. I would suggest replacing the word “downward”(L. 119)

2. L.127-130, I would consider rewriting the sentence with minor changes to make it clearer (e.g. parental communication is not related to high risk of adolescent pregnancy but I believe poor parental communication is)

3. L. 102: reference missing for –“ despite sizable body of literature”

4. L.106, 358 etc. I would consider putting the reference at the end of the sentence or after some words or data from the study but not directly after one word – “According” or “Similarly”. Eg: “According to a study, these persons ……. [18] “OR “According to a study [18], these persons ……. “

5. L. 332,334: I suggest restructuring the last part of the sentence.

Reviewer #3: The statistical analysis is fairly simple compared to the conceptual challenges presented in the study. There is one issue that may require the opinion of a statistician regarding the inclusion of country as a random effect in the model.

The full review comments can be found in an attachment.

6. PLOS authors have the option to publish the peer review history of their article (what does this mean?). If published, this will include your full peer review and any attached files.

Reviewer #1: Yes: RODGERS MAKWINJA

Reviewer #2: Yes: Masna Rai

Reviewer #3: No

---

## [Author Response · Author response to Decision Letter 0]

6 Apr 2020

AUTHOR’S RESPONSE TO REVIEWS

Title: Female adolescents’ reproductive health decision-making capacity and contraceptive use in sub-Saharan Africa: What does the future hold?

Dear Editor and Reviewers,

Thank you for your email dated 9th March 2020 enclosing the reviewer’s comments. On behalf of all authors, I convey our gratitude to you for the critical and constructive review that has led to the massive improvement of our paper entitled “Female adolescents’ reproductive health decision-making capacity and contraceptive use in sub-Saharan Africa: What does the future hold?”. We have carefully reviewed the comments from all the three reviewers and have revised the manuscript accordingly. Our responses are given in a point-by-point manner below.Most of the changes have been indicated in yellow colour. We believe the manuscript has improved substantively and will be published in your reputable journal. 

Version 1: PONE-D-19-29601

Date:4/4/2020

REVIEWER #1:

Comment: In the methodology, the author/authors suggested the use of Bivariate and multivariable analysis which include Pearson chi-square tests and binary logistic regression respectively, however, binary logistic regression model in most cases is misleading in interpreting the complex relationships of this nature. The authors need to demonstrate how they deal with existence of multicollinearity problems which obviously made the model deficiency.

I feel that the problem of multicollinearity cannot be undermined in this study because it can lead the authors into making incorrect conclusion about relationships between responsive and explanatory variables. Therefore, I would suggest that the authors should consider adopting Principle component analysis (PCA) in addition to regression approaches to avoid this problem. The PCA is a multivariate technique which help to understand the underlying data structure and to form a smaller number of uncorrelated new variables. In other words, PCA reduces number of predictors to avoid multicollinearity problem and it is highly recommended due to its ability to identify a small number of derived variables from a larger number of original variables in order to simplify the subsequent analysis of the data.

Response: Before results were interpreted, multicollinearity between explanatory variables was assessed through the variance inflation factors (VIF) at a reference value of 5 (We have attached this as a supplementary file. (page 7)

REVIEWER #2:

I acknowledge the authors for a comprehensive and in terms of language, a well-written paper. The paper was easy to follow. Common, yet important limitations of studies of such types involving sexual and reproductive health (SRH) were detailed in the “Limitation” section. Few comments I have are as follows:

Comment 1. Important things to mention in the last part of the “Introduction” section always are to briefly mention similar studies, what they lacked (the research gaps) and which gaps the current study aims to fulfil. This would make clear for the readers how the study is different from other studies with similar research objective. This important aspect is seen to be missing in the “Introduction” part. An example for it would be: In the material and method section, the authors mention to have followed previous studies (referenced 33 and 34) (Line no. 190). We see the topic of these two studies are similar to the current study, and study of such types are existent, however, the authors neither mention this study in the “Introduction” nor the gaps these studies had and the gaps that the current study attempts to cover. Another similar comment when the authors directly mention other two studies (reference 42, 43) (L.352) in the discussion and compare their results with them.

Response: We have revised the introduction of our manuscript to take into consideration these issues (see Page 3-5). 

Comment 2. Although the study covered many aspects determining the reproductive decision making and contraceptive use, some interesting analysis could have been carried out additionally. I suggest a subgroup analysis grouping the countries with SRH or contraceptive intervention and the countries without. I believe that reproductive decision alone cannot facilitate contraceptive use (To explain, even if one has the reproductive decision making capacity but no availability of contraceptive, he/she would not be able to use contraceptives despite the will. On the other hand, if a person has very easy access to contraceptives, a slight decision making capacity would also be enough for contraceptive use).

Response: Even though sexual and reproductive health interventions are not evenly distributed in sub-Saharan Africa, each country has instituted various policies to tackle contraceptive usage: Please see (Liang, M., Simelane, S., Fillo, G. F., Chalasani, S., Weny, K., Canelos, P. S., ... & Michielsen, K. (2019). The state of adolescent sexual and reproductive health. Journal of Adolescent Health, 65(6), S3-S15.

Melesse, D. Y., Mutua, M. K., Choudhury, A., Wado, Y. D., Faye, C. M., Neal, S., & Boerma, T. (2020). Adolescent sexual and reproductive health in sub-Saharan Africa: who is left behind?. BMJ Global Health, 5(1).; https://www.afro.who.int/health-topics/sexual-and-reproductive-health)

Comment: 3. L: 258-270: I suggest mentioning the CI of the given results. (CI across different countries).

Response: We have presented the results as forest plots and the results are now showing their respective 95% Cis (Please see Fig 1-4). 

Minor comments:

Comment: 1. I would suggest replacing the word “downward”(L. 119)

2. L.127-130, I would consider rewriting the sentence with minor changes to make it clearer (e.g. parental communication is not related to high risk of adolescent pregnancy but I believe poor parental communication is)

Response: We have revised the background based on the recommendations made by Reviewer 3 and all these issues have been addressed (see Page 3-5).

Comment: 3. L. 102: reference missing for –“ despite sizable body of literature”

Response: We have revised the background based on the recommendations made by Reviewer 3 (see Page 3-5).

Comment: 4. L.106, 358 etc. I would consider putting the reference at the end of the sentence or after some words or data from the study but not directly after one word – “According” or “Similarly”. Eg: “According to a study, these persons ……. [18] “OR “According to a study [18], these persons ……. “

5. L. 332,334: I suggest restructuring the last part of the sentence.

Response: We have taken note of this and revised some of these statements in the manuscript.

REVIEWER #3:

The statistical analysis is fairly simple compared to the conceptual challenges presented in the study. There is one issue that may require the opinion of a statistician regarding the inclusion of country as a random effect in the model. Overall the study shines a lens on the important challenge of ensuring adolescents have the power to make decisions about their reproductive health, including their contraceptive use. However, the study is hampered by insufficient nuance in the consideration of both its main explanatory and outcome variables. The paper would be much strengthened by a more critical discussion and reflection on this. My main critiques, described in specific comments below, refer to refining the crude measure of reproductive health decision-making capacity, considering the contraceptive need (rather than just contraceptive use) of adolescents, defining the ‘sexually active’ adolescent population, and expanding the outcome to modern contraception (rather than just any contraception).

Major issues

Introduction

Comment: Overall, the Introduction is very long and could be written in more clear and accessible language. The authors should consider substantially shortening the Introduction and having a clear message for each paragraph. I suggest beginning with a very brief summary on trends/regional prevalence of adolescent contraceptive use prior to the discussion of women’s empowerment conceptualisation and measurement.

Response: We have revised the background taking on board all the suggestions (see Page 3-5).

Methods

Comment: How was the DHS for each country selected? Was only the most recent survey used? Please provide a full list of countries and survey years (also see note in Results).

Response: We mentioned in the methods how each DHS was selected and have provided a Table that contains the countries and their survey years and the sample (See Table 1). 

Comment: Pg 7, line 168: “same questions were posed to all women making it feasible for multi-country 169 study.” This isn’t quite true. While there is a standard DHS questionnaire, some adaptations are made to each survey. Were the countries and surveys selected for inclusion in this analysis based on whether they had certain questions included?

Response: The countries were selected because they had similar variables. “The DHS contain both ‘core’ and optional questionnaires (modules). Core questionnaires cover basic demographic and health content, including marriage, fertility, family planning, reproductive health and child health, whereas additional modules contain special topics, including maternal mortality, men’s survey, anthropometry (height and weight measurement), anaemia blood testing, gender/domestic violence, malaria, maternal mortality, tobacco use, chronic illnesses and other biomarkers” Consequently, each survey is tailored to the needs of a particular country while containing several basic components that are comparable across all countries ( Corsi, Neuman, Finlay & Subramanian, 2012 p.1606). Corsi, D. J., Neuman, M., Finlay, J. E., & Subramanian, S. V. (2012). Demographic and health surveys: a profile. International journal of epidemiology, 41(6), 1602-1613.

Comment: Pg 8, line 172: How was the population of sexually active adolescent girls identified in the datasets? Are these women ‘in union’ or was this based on time since last sexual intercourse? How was the need for contraception considered in the analysis, as some sexually active adolescents may want to become pregnant?

Response: The population for sexually active adolescents were determined by those who indicated they have ever had sex and mentioned their age at first sex as used in previous studies (Appiah et al 2020). Please we acknowledge the need for contraception as a limitation in our study. (see page 26)

Comment: Pg 8, line 178: What methods are included in modern methods? Did the authors consider examining the outcome of modern method of contraception, rather than just any method of contraception? Why was modern method of contraception not used as a (secondary) outcome, particularly as using a folkloric method could reflect a desire to avoid pregnancy but an inability to access a more effective method?

Response: The modern methods included female sterilization, male sterilization, intrauterine contraceptive device (IUD), contraceptive injection, contraceptive implants (Norplant), contraceptive pill, condoms, emergency contraception, standard day method (SDM), vaginal methods (foam, jelly, suppository), lactational amenorrhea method (LAM), country-specific modern methods and respondent-mentioned other modern contraceptive methods (including cervical cap, contraceptive sponge, and others). Aside examining contraceptive use in general, we also examined the factors associated with modern contraceptive use (secondary outcome variable) (see Table 3 ). 

Comment: For the explanatory variable of ‘reproductive health decision-making capacity’, how were discordant answers between the two component variables handled? Did the respondent need to have answered ‘yes’ to both refuse partners for sex and condom use in order to be considered to have ‘reproductive health decision-making capacity’? It seems likely that women using another contraceptive method might find it strange or difficult to request that their partner, particularly in longer-term or more serious relationships, use a condom (double protection). Did the authors conduct further analysis of how each component of the binary capacity indicator performed separately in relation to the study outcome? The DHS also often asks questions on experience of violence and/or coercive sex. Did the authors consider other potential constructs to include in the indicator? Further discussion of the construction and limitations of this variable to capture the extremely complex concept of ‘reproductive health decision-making capacity’ is needed.

Response: We followed the measurement of ‘reproductive health decision-making capacity’ by previous studies:

Darteh EK, Doku DT, Esia-Donkoh K. Reproductive health decision-making among Ghanaian women. Reproductive health. 2014;11(1):23 

Darteh, E. K. M., Dickson, K. S., & Doku, D. T. (2019). Women’s reproductive health decision-making: A multi-country analysis of demographic and health surveys in sub-Saharan Africa. PloS one, 14(1)

Seidu AA, Ahinkorah BO, Agbemavi W, Amu H, Bonsu F. Reproductive health decision-making capacity and pregnancy termination among Ghanaian women: Analysis of the 2014 Ghana demographic and health survey. Journal of Public Health. 2019:1-0.

We have acknowledged limitation in the construction of ‘reproductive health decision-making capacity’ in our manuscript and recommend further studies to look at these limitations to refine its measurement (page 26.) 

Comment: Pg 9, line 212-221: When combining surveys in multi-country studies to produce an overall result for sub-Saharan Africa (or any region), it is often necessary to weight country-specific estimates by the country’s population. This is so that the result from a very large country like Nigeria has a greater weight in the combined effect analysis than a very small country such as Comoros. (Previous multi-country studies explaining how to apply population weights:https://doi.org/10.1016/j.jadohealth.2017.09.013 or https://doi.org/10.1111/tmi.12597) Did the authors have a reason for not applying population weights, in addition the individual survey weights standard in all DHS analysis, in the calculation of regional statistics? Finally, I defer to a statistician, but wonder whether country is more appropriate as a random effect rather than a covariate in the models.

Response: We applied both population weight and sample weights (see page 10) .Again a test of heterogeneity of the DHS data obtained for the different countries showed a high level of inconsistency (I2 > 50%) thereby warranting the use of a random effect model in all the meta-analysis (see Figures 1, 2, 3 and 4), but at the regression models as used in previous studies, country was considered a covariate: 

Darteh, E. K. M., Dickson, K. S., & Doku, D. T. (2019). Women’s reproductive health decision-making: A multi-country analysis of demographic and health surveys in sub-Saharan Africa. PloS one, 14(1); 

Ameyaw, E. K., Budu, E., Sambah, F., Baatiema, L., Appiah, F., Seidu, A. A., & Ahinkorah, B. O. (2019). Prevalence and determinants of unintended pregnancy in sub-Saharan Africa: A multi-country analysis of demographic and health surveys. PloS one, 14(8).

Results

Comment: Please provide a summary table of country, survey year, sample size of sexually active adolescents and proportion of sexually active adolescents out of all adolescents. Please also provide a summary table of the distribution of covariates. These tables can be either part of the main paper or in supplementary materials.

Response: This has been provided. See Table 1.

Comment: Figure 2: As the reproductive decision-making capacity is a simplistic measure, it would be helpful to provide estimates for each of the two components, either in the main paper or in supplementary materials.

Response: We have provided the estimates in the forest plots. See Figure 2-4

Comment: Pg 16, lines 299-300: This links to a question in the Methods section about the need for contraception. Married adolescents may well be desiring a pregnancy so have no need for contraception.

Response: We have acknowledged this as a limitation in the paper (page 26.)

Discussion

Comment: Overall the Discussion section is far too long and would do well to more succinctly summarise the key findings and their implications. The limitations section is insufficient in examining the severe limitations of the indicators used for both explanatory and outcome variables. ‘Reproductive health decision-making capacity’ is a complex construct reduced to a binary variable based on two components. The outcome of contraceptive use is likewise insufficiently nuanced to consider whether the adolescent has a need to avoid pregnancy.

Response: We have revised the discussion and acknowledged the limitations in measurement of both the explanatory and outcome variables( See page 21-27). 

Comment: Pg 21, lines 374-381: This is overly speculative and not backed up either by the results or this study or properly cited. Earlier sexual debut may itself be a marker of ‘empowerment’ or lack thereof, particularly if the first sex was coerced. The DHS has further questions that could have allowed a more nuanced of this construct. Second, as the population was sexually active adolescents age 15-19, many would only very recently have had their sexual debut. Why did the authors not use more nuanced categories for age of sexual debut, particularly since all the 15- year-olds in the analysis, by definition, would fall into the <16 years at sexual debut category. What is sexual debut as the binary variable here actually telling us?

Response: We have revised the discussion. 

Comment: Pg 22, line 403-404: Here, and implied elsewhere in the paper, is the assertion that increasing education or empowering adolescent girls may be an effective tool for increasing adolescent contraceptive use, instead of a goal worthwhile in itself. For example: “It is imperative to strengthen existing efforts in SSA on contraceptives use among adolescent by empowering adolescents to take reproductive health decisions” (line 499-500). The authors would do well to engage more carefully in the discussion of empowerment as a means to an end (contraceptive use) rather than simply the end itself. In particular due to the paper’s outcome of contraceptive use, rather than met need for contraception, the implication is that all adolescents should be using contraception, rather than all adolescents having freedom and power to choose to use contraception if they wish.

Response: We have revised this section. See page 24)

Minor issues

Comment: Suggest a careful read as typos were noted.

Response: We have reviewed and edited the manuscript extensively. The manuscript has also been reviewed by a native English speaker.

Introduction

Comment: Some stylistic edits needed, such as in page 4, line 94, to say ‘For example, Ann Blanc [15] opined…’ rather than listing the numeric reference only.

Response: These issues have been rectified. 

Methods

Comment: Discussion of data source and sampling could be shortened as readers can reference the individual surveys for details of the specific sampling procedures.

Response: This section has been shortened. 

Comment: Pg 9, line 197: Is education level based on highest level of completed education? How were these grouped into categories? As some younger adolescents may still be in education, how did the authors consider this in the analysis?

Response: Thank you. This is on highest level of education completed

Comment: Pg 9, line 197: Is country of origin the same as the survey country? Or does this refer to DHS questions about migration?

Response: This is referring to surveyed country. This has been clarified in the methods section of the manuscript. 

Comment: Pg 9, line 199: “based on their significant association with the outcome variable–contraceptives use” Was there a specific threshold used?

Response: The threshold was that all those studies used nationally-representative survey data.

Comment: Pg 9, line 203: How did the authors code the occupation status of respondents still in education (students)?

Response: Education was captured based on highest level completed. 

Results

Comment: Figure 1: How have the countries been sorted in some way for this bar chart? It might make more sense to sort alphabetically.

Response: This has been presented with forest plot (see Figure 1-4). 

Comment: Table 1: This table could be simplified by showing a total n for each row (instead of a sample size for ‘no’ and for ‘yes’). Only the percentage of ‘yes’ is needed (as the ‘no’ for any contraceptive use is just the corresponding fraction). Please provide confidence intervals.

Response: This has been presented with forest plot (see Figure 1-4). 

Comment: Pg 16, line 290: Is there a reason the comparison or reference country is Angola?

Response: We have revised this. We have used the country with the lowest proportion (Zambia) as the reference category. 

Discussion

Comment: Pg 19, line 323: Could the very different estimate be related to the crude measure of reproductive decision-making capacity measure used in this study?

Response: This section has been revised. 

Comment: Pg 23, line 430: “Female adolescents’ ability to make reproductive health decisions was associated with contraceptive use, which in turn could be connected with perceived unmet need.” This point about contraceptive use connected with perceived unmet need is confusing.

Response: This statement has been revised.

---

## [Decision Letter · Decision Letter 1]

20 May 2020

PONE-D-19-29601R1

Female adolescents’ reproductive health decision-making capacity and contraceptive use in sub-Saharan Africa: What does the future hold?

PLOS ONE

Dear Mr Seidu,

Thank you for submitting your manuscript to PLOS ONE. After careful consideration, we feel that it has merit but does not fully meet PLOS ONE’s publication criteria as it currently stands. Therefore, we invite you to submit a revised version of the manuscript that addresses the points raised during the review process.

You will see that requested revisions are indeed quite minor. Two reviewers have asked you to edit, and a third reviewer has detailed additional revisions that will also clarify the manuscript. 

We would appreciate receiving your revised manuscript by Jul 04 2020 11:59PM. To enhance the reproducibility of your results, we recommend that if applicable you deposit your laboratory protocols in protocols.io, where a protocol can be assigned its own identifier (DOI) such that it can be cited independently in the future. For instructions see: http://journals.plos.org/plosone/s/submission-guidelines#loc-laboratory-protocols

We look forward to receiving your revised manuscript.

Kind regards,

Emily Vala-Haynes

Academic Editor

PLOS ONE

Reviewers' comments:

Reviewer's Responses to Questions

**Comments to the Author**

1. If the authors have adequately addressed your comments raised in a previous round of review and you feel that this manuscript is now acceptable for publication, you may indicate that here to bypass the “Comments to the Author” section, enter your conflict of interest statement in the “Confidential to Editor” section, and submit your "Accept" recommendation.

Reviewer #1: All comments have been addressed

Reviewer #2: (No Response)

Reviewer #3: (No Response)

2. Is the manuscript technically sound, and do the data support the conclusions?

Reviewer #1: Yes

Reviewer #2: Yes

Reviewer #3: Yes

3. Has the statistical analysis been performed appropriately and rigorously? 

Reviewer #1: Yes

Reviewer #2: Yes

Reviewer #3: Yes

4. Have the authors made all data underlying the findings in their manuscript fully available?

Reviewer #1: Yes

Reviewer #2: Yes

Reviewer #3: Yes

5. Is the manuscript presented in an intelligible fashion and written in standard English?

Reviewer #1: Yes

Reviewer #2: Yes

Reviewer #3: Yes

6. Review Comments to the Author

Reviewer #1: The paper can be accepted however, the authors need to critically attend few grammatical and sentence structures

Reviewer #2: The authors have addresses my comments.

Although I would consider revising lines 102 and 103 in the introductions - the use of lines "over-exaggerated or underreported BY THE AUTHORS" to something like "the study does not give a clear picture" that sounds more subtle or "...there are chances of under and overreporting in the study", but not say "by the authors".

Reviewer #3: Thank you to the authors for the changes made in the revised manuscript. In particular the Results are presented much more clearly, and this is greatly appreciated. I still have some concerns about the Methods used and the importance of detailing them in the paper. I’ve noted the major questions and suggestions below. Finally, I continue to disagree with the authors’ decision to use contraceptive use, rather than met need for contraception, as the outcome in this analysis because this is out-of-step with current trends in the family planning literature which recognises the importance of women’s reproductive agency rather than simply achieving high contraceptive prevalence. However, if contraceptive use is the outcome the authors wish to use, perhaps this can be justified more explicitly in the Introduction or Methods.

Line 139: Please clarify (in the text of the paper, not just the response to reviewer comments) how ‘sexually active adolescents’ was defined in the analysis. Though the authors say that they used the same classification as Appiah et al. 2020, this paper isn’t cited. I also note that just because an adolescent woman has ever had sex, it does not mean that she is sexually active at the time of the survey. Adolescents who have had their sexual debut but have not had sex in the past year can be a substantial proportion of so-called ‘sexually active’ adolescents (a phenomenon sometimes called secondary abstinence). There is a clear space to have more nuance in the definition of ‘sexually active’, such as those used in these papers:

https://www.dhsprogram.com/pubs/pdf/CR29/CR29.pdf

https://doi.org/10.1016/S2214-109X(20)30060-7

https://doi.org/10.1017/S002193201900083X

Or the authors should justify why they have used only ‘has ever had sex’ in defining the population for their analysis.

Table 1: This needs better labelling to clarify what ‘Sample (n)’ and ‘Sample (%)’ refer to. It would also be helpful to include in this table the total number of women age 15-19 sampled in each country’s survey, as the proportion of sexually active adolescents in each country is likely to vary considerably. Also for the ‘Sample (%)’, is this population weighted?

Line 177: Marital status should be listed as one of the other explanatory variables. Were all the women in the analysis ‘in union’, that is married or cohabitating? If so, then was this a component of selection for the sub-population of ‘sexually active adolescents’ for the analysis? It seems possible that an unmarried adolescent could be sexually active but not live with her sexual partner – was this accounted for and if so, how?

Line 192: How were population weights added? Did the authors use each country’s population of women age 15-49 for the median survey year for all 32 surveys? Or the population for each country for the specific survey year? This needs to be explained.

Pg 9: How was missing data handled? It appears that all 11,474 women in the analysis had data for all variables, but what was the extent of missingness? Were there any variables (explanatory and outcome) with a substantial proportion of women missing this information and thus excluded from the analysis?

Discussion: One major factor that is under-discussed is the role of marital status. Marital status is important in many contexts in the acceptability of sexual activity, the desire to become pregnant and the ease of accessing health services, including for family planning. Early marriage likewise has important implications for empowerment (for both directions of effect, as marriage can also confer social status). I would like to see this explored in the discussion of the results.

Line 461: What does it mean to be ‘validated’? Just because questions are used often in DHS does not automatically mean they are valid! In fact, widely used DHS contraception questions have been shown to be interpreted in rather different ways than intended:

https://www.dhsprogram.com/publications/publication-qrs20-qualitative-research-studies.cfm

This does not make your study “valid” or “generalisable” to other adolescents in SSA.

7. PLOS authors have the option to publish the peer review history of their article (what does this mean?). If published, this will include your full peer review and any attached files.

Reviewer #1: Yes: Rodgers Makwinja

Reviewer #2: Yes: Masna Rai

Reviewer #3: Yes: Emma Radovich

---

## [Author Response · Author response to Decision Letter 1]

10 Jun 2020

AUTHOR’S RESPONSE TO REVIEWS

Title: Female adolescents’ reproductive health decision-making capacity and contraceptive use in sub-Saharan Africa: What does the future hold?

Dear Editor and Reviewers,

Thank you for your email dated 20th May 2020 enclosing the reviewer’s comments. On behalf of all authors, I convey our gratitude to you for the critical and constructive review that has led to the massive improvement of our paper entitled “Female adolescents’ reproductive health decision-making capacity and contraceptive use in Sub-Saharan Africa: What does the future hold?”. We have carefully reviewed the comments from all the three reviewers and have revised the manuscript accordingly. Our responses are given in a point-by-point manner below. Most of the changes have been indicated in yellow colour. We believe the manuscript has improved substantively and will be published in your reputable journal. 

Version 2: PONE-D-19-29601R1

Date:2/6/2020

REVIEWER #1

1. Comment: The paper can be accepted however, the authors need to critically attend few grammatical and sentence structures

2. Response: Please we have given the manuscript to a proof reader who has helped us address the few grammatical and sentence structures errors.

REVIEWER #2

3. Comment: The authors have addresses my comments.

Response: Thank you for your time. 

4. Comment: Although I would consider revising lines 102 and 103 in the introductions - the use of lines "over-exaggerated or underreported BY THE AUTHORS" to something like "the study does not give a clear picture" that sounds more subtle or "...there are chances of under and overreporting in the study", but not say "by the authors".

Response: This has been revised to read “Hence, there are chances of under and over-reporting in the study(see page 5). 

REVIEWER #3

5. Comment: Thank you to the authors for the changes made in the revised manuscript. In particular the Results are presented much more clearly, and this is greatly appreciated. I still have some concerns about the Methods used and the importance of detailing them in the paper. I’ve noted the major questions and suggestions below. Finally, I continue to disagree with the authors’ decision to use contraceptive use, rather than met need for contraception, as the outcome in this analysis because this is out-of-step with current trends in the family planning literature which recognises the importance of women’s reproductive agency rather than simply achieving high contraceptive prevalence. However, if contraceptive use is the outcome the authors wish to use, perhaps this can be justified more explicitly in the Introduction or Methods.

Response: Thanks for your time and the appreciation of our effort. We have addressed the comments raised at the various sections of the manuscript. 

6. Comment: Line 139: Please clarify (in the text of the paper, not just the response to reviewer comments) how ‘sexually active adolescents’ was defined in the analysis. Though the authors say that they used the same classification as Appiah et al. 2020, this paper isn’t cited. I also note that just because an adolescent woman has ever had sex, it does not mean that she is sexually active at the time of the survey. Adolescents who have had their sexual debut but have not had sex in the past year can be a substantial proportion of so-called ‘sexually active’ adolescents (a phenomenon sometimes called secondary abstinence). There is a clear space to have more nuance in the definition of ‘sexually active’, such as those used in these papers: https://www.dhsprogram.com/pubs/pdf/CR29/CR29.pdf

https://doi.org/10.1016/S2214-109X(20)30060-7

https://doi.org/10.1017/S002193201900083X

Or the authors should justify why they have used only ‘has ever had sex’ in defining the population for their analysis.

Response: Thank you for your valid comment. The focus of the study was adolescents in unions (cohabiting or married). The reproductive health decision making variable focuses on only those married or cohabiting. This has been clarified in the manuscript (see page 5 ).

7. Comment: Table 1: This needs better labelling to clarify what ‘Sample (n)’ and ‘Sample (%)’ refer to. It would also be helpful to include in this table the total number of women age 15-19 sampled in each country’s survey, as the proportion of sexually active adolescents in each country is likely to vary considerably. Also for the ‘Sample (%)’, is this population weighted?

Response: Table 1 has been clarified and well labelled (see Table 1 page 7). 

8. Comment: Line 177: Marital status should be listed as one of the other explanatory variables. Were all the women in the analysis ‘in union’, that is married or cohabitating? If so, then was this a component of selection for the sub-population of ‘sexually active adolescents’ for the analysis? It seems possible that an unmarried adolescent could be sexually active but not live with her sexual partner – was this accounted for and if so, how?

Response: Thank you for your comment. Marital status has been listed as one of the explanatory variables (see page 9). Again, we used adolescents in sexual unions. This means that those who are sexually active but were neither married nor cohabiting were excluded. This was influenced by the key independent variable (reproductive health decision making) which focuses on only women in sexual unions (see page 6).

9. Comment: Line 192: How were population weights added? Did the authors use each country’s population of women age 15-49 for the median survey year for all 32 surveys? Or the population for each country for the specific survey year? This needs to be explained.

Response: We have clarified how weighting was done (see page 10). Again, the population for the study was adolescents aged 15-19 on whom data was collected during the recent survey for each of the countries (see page 6)

10. Comment: Pg 9: How was missing data handled? It appears that all 11,474 women in the analysis had data for all variables, but what was the extent of missingness? Were there any variables (explanatory and outcome) with a substantial proportion of women missing this information and thus excluded from the analysis?

Response: We treated missing values by using complete cases for our analysis. Moreover, no variable was excluded from the analysis because it had a substantial proportion of missing data (see page 10). 

11. Comment: Discussion: One major factor that is under-discussed is the role of marital status. Marital status is important in many contexts in the acceptability of sexual activity, the desire to become pregnant and the ease of accessing health services, including for family planning. Early marriage likewise has important implications for empowerment (for both directions of effect, as marriage can also confer social status). I would like to see this explored in the discussion of the results.

Response: The discussion on marital status has been added. See page 25.

12. Comment: Line 461: What does it mean to be ‘validated’? Just because questions are used often in DHS does not automatically mean they are valid! In fact, widely used DHS contraception questions have been shown to be interpreted in rather different ways than intended:

https://www.dhsprogram.com/publications/publication-qrs20-qualitative-research-studies.cfm

This does not make your study “valid” or “generalisable” to other adolescents in SSA.

Response: We have revised this section of the manuscript (see page

---

## [Editor Report · Decision Letter 2]

19 Jun 2020

Female adolescents’ reproductive health decision-making capacity and contraceptive use in sub-Saharan Africa: What does the future hold?

PONE-D-19-29601R2

Dear Dr. Seidu,

We’re pleased to inform you that your manuscript has been judged scientifically suitable for publication and will be formally accepted for publication once it meets all outstanding technical requirements.

Kind regards,

Olanrewaju Oladimeji, Ph.D., MB; BS

Academic Editor

PLOS ONE

Additional Editor Comments (optional):

Authors have addressed the concerns raised by the reviewers.
---

## [Editor Report · Acceptance letter]

30 Jun 2020

PONE-D-19-29601R2 

Female adolescents’ reproductive health decision-making capacity and contraceptive use in sub-Saharan Africa: What does the future hold? 

Dear Dr. Seidu:

I'm pleased to inform you that your manuscript has been deemed suitable for publication in PLOS ONE. Congratulations! Your manuscript is now with our production department. 

Kind regards, 

on behalf of

Dr. Olanrewaju Oladimeji 

Academic Editor

PLOS ONE